# Evaluating the motional time scales contributing to averaged anisotropic interactions in MAS solid-state NMR

Kathrin Aebischer[1], Lea Marie Becker[2], Paul Schanda[2], and Matthias Ernst[1]

[1]Department of Chemistry and Applied Biosciences, ETH Zürich, Vladimir-Prelog-Weg 2, 8093 Zürich, Switzerland
[2]Institute of Science and Technology Austria (ISTA), Am Campus 1, 3400 Klosterneuburg, Austria.

**Correspondence:** Matthias Ernst (maer@ethz.ch)

**Abstract.**

Dynamic processes in molecules can occur on a large range of time scales, and it is important to understand which time scales of motion contribute to different parameters used in dynamics measurements. For spin relaxation, this can easily be understood from the sampling frequencies of the spectral-density function by different relaxation-rate constants. In addition to data from relaxation measurements, determining dynamically-averaged anisotropic interactions in magic-angle spinning (MAS) solid-state NMR allows better quantification of the amplitude of molecular motion. For partially averaged anisotropic interactions, the relevant time scales of motion are not so clearly defined. Whether the averaging depends on the experimental methods (e.g., pulse sequences) or conditions (e.g., MAS frequency, magnitude of anisotropic interaction, radio-frequency-field amplitudes) is not fully understood. To investigate these questions, we performed numerical simulations of dynamic systems based on the stochastic Liouville equation using several experiments for recoupling the dipolar-coupling, chemical-shift anisotropy or quadrupolar coupling. As described in the literature, the transition between slow motion, where parameters characterizing the anisotropic interaction are not averaged, and fast motion, where the tensors are averaged leading to a scaled anisotropic quantity, occurs over a window of motional rate constants that depends mainly on the strength of the interaction. This transition region can span two orders of magnitude in exchange-rate constants (typically in the µs range) but depends only marginally on the employed recoupling scheme or sample spinning frequency. The transition region often coincides with a fast relaxation of coherences making precise quantitative measurements difficult. Residual couplings in off-magic-angle experiments, however, average over longer time scales of motion. While in principle one may gain information on the time scales of motion from the transition area, extracting such information is hampered by low signal-to-noise ratio in experimental spectra due to fast relaxation that occurs in the same region.

# 1 Introduction

Nuclear magnetic resonance (NMR) spectroscopy is unique in its ability to probe molecular motions with a resolution of individual atoms or bonds, and allows quantification of the amplitudes and time scales of the motional processes. In magic-angle spinning (MAS) solid-state NMR, two types of approaches are widely used to probe dynamic processes. One class of experiments measures nuclear spin-relaxation rate constants, which are sensitive to the local fluctuating magnetic fields generated by anisotropic interactions, i.e., the dipolar couplings to spatially close spins, the chemical-shift anisotropy (CSA) of the nucleus, or (for spins with $I > 1/2$) the quadrupolar coupling (Lewandowski, 2013; Krushelnitsky et al., 2013; Lamley and Lewandowski, 2016; Schanda and Ernst, 2016). Relaxation-rate constants vary in their sensitivities to different time scales of motion by sampling the spectral-density function at different frequencies. For example, relaxation of a $^{15}N_z$ spin state ($T_1$ relaxation) due to the $^1H$-$^{15}N$ dipolar coupling is fastest if the motion occurs on a nanosecond time scale, while relaxation of $^{15}N_{x,y}$ coherence in the presence of a spin-lock radiofrequency field ($T_{1\rho}$ relaxation) is fastest when it takes place on a µs time scale (Schanda and Ernst, 2016). Spin relaxation measurements can, therefore, be used to extract the amplitudes and time scales of the motion. However, disentangling amplitudes and time scales is difficult and the solution may be ambiguous if multiple motions on different time scales are present (Zumpfe and Smith, 2021). For instance, using only $^{15}N$ $T_1$ and $T_{1\rho}$ relaxation times in proteins leads to a systematic underestimation of the amplitude of motion (Haller and Schanda, 2013; Lamley et al., 2015).

The second type of approach measures how anisotropic interactions, e.g., dipolar couplings, chemical-shift tensors or quadrupolar couplings, are averaged by motion (Brüschweiler, 1998; Hou et al., 2012; Yan et al., 2013; Watt and Rienstra, 2014; Schanda and Ernst, 2016). The orientation dependence of these interactions leads to motional averaging, resulting in an interaction that is the average over all the sampled conformational states. As these second-rank tensors are traceless, the time-averaged interaction strength becomes zero in the limiting case where all orientations in space are sampled with equal probability (isotropic motion). Thus, in the presence of overall tumbling, i.e., in isotropic solution, anisotropic interactions are averaged to zero, and provide no direct information about dynamics. The interactions are, however, the source of relaxation by generating fluctuating local fields. Restricted motion without overall tumbling results in a reduced magnitude of the tensor. Depending on the symmetry of the motional process, the symmetry of the tensor can change under dynamic averaging. For motional processes with at least three-fold symmetry, the tensor is characterized by a single parameter, the tensor anisotropy, $\delta$. For a dipolar coupling the averaged and thus reduced anisotropy, $\delta_{IS}^{red}$, and the ratio of this value over the tensor anisotropy for the rigid-limit case, $\delta_{IS}^{rigid}$, report on the amplitude of the motion. It is often expressed as the dipolar order parameter, $S_{IS} = \delta_{IS}^{red}/\delta_{IS}^{rigid}$. The rigid-limit tensor parameters are well known for dipolar couplings, where the anisotropy $\delta_{IS}^{rigid}$ only depends on the distance between the spins and their gyromagnetic ratios, and the tensor asymmetry $\eta$ is zero. For chemical-shift anisotropy and quadrupolar couplings, obtaining the rigid-limit value is only possible from quantum-chemical calculations or by freezing out the averaging process. The first approach can be very demanding for larger molecules while the latter one is experimentally complex due to the loss of resolution in low-temperature MAS NMR experiments (Concistrè et al., 2014). In

the general case, dynamically averaged anisotropic interactions can become asymmetric ($\eta \neq 0$), even if the rigid-limit tensor is axially symmetric. One can exploit this feature to reveal motions with no (or low) symmetry, such as aromatic ring flips or side-chain motions in proteins from determining not only the residual anisotropy $\delta_{\mathrm{IS}}^{\mathrm{red}}$ but also the residual asymmetry parameter $\eta_{\mathrm{IS}}^{\mathrm{red}}$ (Hong, 2007; Schanda et al., 2011a; Gauto et al., 2019).

The efficiency of the averaging process depends on the time scale of the underlying motion: in the limiting case of very slow motion, the rigid-limit interaction strength is observed, while in the opposite extreme of very fast motion the observed interaction strength reflects the population-weighted average over the sampled conformations. Although it is often stated that the dynamic averaging is effective over all motions with a time scale shorter than the inverse of the interaction strength, e.g., tens of µs for a typical one-bond $^{1}$H-$^{13}$C dipolar coupling (Chevelkov et al., 2023), the exact time scale of the fast motion limit and whether it depends on the way the interaction is measured, is not fully understood. This is, however, an important question since it defines which time scales are characterized by the measured order parameter and has several important implications. Firstly, this knowledge allows the assignment of a lower limit on the time scale of the underlying motional averaging processes observed in experiments and is crucial when measurements of dynamically averaged anisotropic interactions are used in combination with relaxation data. Such a combination is invaluable, since the order parameter, $S$, obtained from the averaging of anisotropic interactions, greatly improves the fit of motional time scales from relaxation data. For example, in the commonly employed detectors approach (Smith et al., 2018; Zumpfe and Smith, 2021) used for fitting relaxation data, dynamics is described by the amplitudes of motion in different time windows. Different relaxation rate constants exhibit varying sensitivities across distinct windows. The total motional amplitude, composed of the amplitudes within each of these time windows, is conveniently limited to the one derived from averaged anisotropic interactions (mostly from dipolar couplings, $1 - S_{\mathrm{IS}}^2$). However, for this approach to be rigorous, one needs to make sure that all time windows used in the detectors are indeed "seen" by the averaged anisotropic interaction. Furthermore, understanding the time scales over which motional averaging occurs can provide information on the time scale of the underlying motion from different experimental measurements of anisotropic interactions. For example, if the same parameter, such as the dipolar-coupling derived order parameter, can be measured by different experiments that involve averaging over different time windows, any disparities in the observed order parameter would indicate motion on time scales detected by one experiment but not the other. If different approaches were to average over different time scales, measuring the same tensor with a variety of methods may provide information on the time scale of motion.

In between the fast- and the slow-motion regime, we find an area which is characterized by intermediate motional time scales in the millisecond to microsecond range. Such motions lead to a fast coherence decay due to rapid $T_2$ relaxation. The details of the transverse relaxation depend on the MAS frequency (Schanda and Ernst, 2016) and on the radiofrequency (rf) irradiation on all involved nuclei. Fast transverse relaxation is most detrimental for the determination of anisotropic interactions through dephasing experiments where an overdamped oscillation may be obtained (deAzevedo et al., 2008). In principle, such dephasing experiments can be relaxation compensated by a reference experiment as is done in the REDOR experiment (Gullion and

Schaefer, 1989; Gullion, 1998). However, care has to be taken to ensure that different rf irradiation schemes in the reference experiment do not lead to different relaxation behavior. For experiments measuring dipolar couplings in a polarization-transfer experiment, mainly the transfer efficiency is affected (Nowacka et al., 2013; Aebischer and Ernst, 2024) but special methods like the Anderson-Weiss formalism (Hirschinger, 2006, 2008) allow the extraction of dipolar couplings (Cobo et al., 2009). In experiments where spinning side bands are observed, differential line broadening of the various side bands is observed (Suwelack et al., 1980; Schmidt et al., 1986; Schmidt and Vega, 1987; Long et al., 1994). Motion on the intermediate time scale does not only affect the recoupling step of the experiment where the anisotropic interactions are measured but also the detection periods as well as all polarization-transfer steps leading to a general attenuation of the affected signals. In cases where the amplitude of motion in the intermediate range is large, this can even lead to entire molecular segments remaining invisible in spectra (Callon et al., 2022). Determining tensor parameters for motion on intermediate time scales, therefore, requires careful consideration of the effects of relaxation during different stages of the experiment. Numerical simulations of exchange on the intermediate time scale under MAS and recoupling sequences have been used to characterize the effects of exchange in this regime using a simplified stochastic Liouville approach (Saalwächter and Fischbach, 2002).

In MAS solid-state NMR experiments, measuring anisotropic interactions usually requires the use of a recoupling sequence since second-rank interactions are averaged out to first order by the MAS. Over the years, many different experiments have been reported for measuring dipolar couplings, including cross-polarization (CP) variants (Lorieau and McDermott, 2006; Chevelkov et al., 2009; Paluch et al., 2013, 2015; Nishiyama et al., 2016; Chevelkov et al., 2023), DIPSHIFT (Munowitz et al., 1981; Jain et al., 2019b), REDOR (Gullion and Schaefer, 1989; Gullion, 1998; Schanda et al., 2010; Jain et al., 2019a) and R sequences (Zhao et al., 2001; Levitt, 2007; Hou et al., 2011) that can also be used to recouple the CSA. Moreover, quadrupolar couplings can be measured under MAS to gain insight into dynamics (Shi and Rienstra, 2016; Akbey, 2022, 2023). In these (recoupling) experiments, the observed oscillation frequency that provides information on the tensor characterizing the anisotropic interaction depends on the type of experiment used, and on the exact parameters (e.g., radiofrequency field strengths and timing) of a given technique. Whether the dynamic averaging also depends on these experimental details has not been analysed systematically. In our study, we use isolated two-spin systems and will not discuss the sensitivity of the various methods to pulse imperfection or to larger spin systems (Schanda et al., 2011b; Asami and Reif, 2017).

In this work, we use numerical simulations based on the stochastic Liouville equation (Kubo, 1963; Vega and Fiat, 1975; Moro and Freed, 1980; Abergel and Palmer, 2003) to investigate the averaging of anisotropic interactions by dynamics over a large range of time scales. We study the dynamic averaging of the dipolar coupling, the chemical-shift anisotropy and the quadrupolar coupling under different experimental conditions. By examining the dependence of the observed tensor parameters on the experimental scheme employed, the size of the rigid-limit tensor and the MAS frequency we provide a quantitative understanding of the motional time scales constituting the fast motion limit. This allows us to characterize which parameters determine the range of motional processes that are seen by the partially averaged anisotropic interactions.

## 2 Methods

Numerical simulations of dipolar and CSA recoupling as well as quadrupolar spectra under MAS were performed using the GAMMA spin simulation environment (Smith et al., 1994). Restricted molecular motion was modeled by a three-site jump process corresponding to a rotation around a $C_3$ symmetry axis (see Fig. 1a). The discrete states used in the jump model only differ in the orientation of the tensors characterizing the anisotropic interactions of interest (dipolar coupling, CSA, quadrupolar coupling). The simulations are based on the stochastic Liouville equation (Kubo, 1963; Vega and Fiat, 1975; Moro and Freed, 1980; Abergel and Palmer, 2003) and are performed in the composite Liouville space of the three states (see Fig. 1b for a schematic depiction of the resulting Liouvillian). A similar approach, where commutation between the exchange operator and the quantum-mechanical Liouvillian was assumed to simplify the calculations, has been used in the characterization of exchange in the intermediate regime combined with MAS and recoupling sequences (Saalwächter and Fischbach, 2002). The dynamic process is included in the simulations through the addition of an exchange super operator. Exchange-rate constants between $1\,\mathrm{s}^{-1}$ and $5\cdot10^{11}\,\mathrm{s}^{-1}$ were simulated (values of 1, 2 and 5 per decade) and a symmetric exchange process and, thus, equal populations of all states assumed (skewed populations would reduce the symmetry of the jump process). The correlation time of this three-site jump process is related to the exchange-rate constant as $\tau_{\mathrm{ex}} = \frac{1}{3k_{\mathrm{ex}}}$. To simplify the data evaluation, only axially-symmetric tensors were considered, i.e., tensors for which the asymmetry parameter $\eta = 0$. Fast exchange leads to the alignment of the scaled anisotropic interaction tensor along the symmetry axis of the three-site jump process. Therefore, the tensor has zero asymmetry both in the static and in the dynamic case and the scaling can be characterized by a single order parameter.

For dipolar recoupling, heteronuclear I-S two-spin spin-1/2 systems with different dipolar-coupling strengths, characterized by the anisotropy of the dipolar coupling tensor $\delta_{\mathrm{IS}}$, were simulated. The anistropy is defined as $\delta_{\mathrm{IS}} = -2\frac{\mu_0}{4\pi}\frac{\hbar\gamma_{\mathrm{I}}\gamma_{\mathrm{S}}}{r_{\mathrm{IS}}^3}$, where $\gamma_{\mathrm{I/S}}$ corresponds to the gyromagnetic ratio of spins I and S respectively, $r_{\mathrm{IS}}$ to the internuclear distance, $\mu_0$ is the permeability of vacuum and $\hbar$ the reduced Planck constant. In these simulations, isotropic and anisotropic chemical shifts, as well as $J$ couplings were neglected. In the case of the CSA recoupling simulations, a one-spin spin-1/2 system was simulated with changing CSA, characterized by the tensor anisotropy $\delta_{\mathrm{CSA}}$, while the isotropic chemical shift and the tensor asymmetry $\eta$ were set to zero. Spectra of deuterium ($^2$H, spin-1 nucleus) were simulated to study the effect of dynamics on quadrupolar nuclei (one-spin system). First- and second-order quadrupolar interactions were taken into account and simulations performed at a static magnetic field of 18.7 T corresponding to a proton Larmor frequency of 800 MHz. Based on literature values (Shi and Rienstra, 2016; Akbey, 2023), a quadrupolar coupling of $C_{\mathrm{qcc}} = 160$ kHz corresponding to an anisotropy of the quadrupolar coupling tensor of $\delta_{\mathrm{Q}}/(2\pi) = C_{\mathrm{qcc}}/(2I(2I-1)) = 80$ kHz was used. The quadrupolar tensor was assumed to be axially symmetric. All simulations were performed in the usual Zeeman rotating frame. Simultaneous averaging over all three powder angles was achieved according to the ZCW scheme (Cheng et al., 1973) and 538 to 10000 crystallite orientations used. For simulations of quadrupoles, 10000 crystallites were required to ensure sufficient $\gamma$-angle averaging and avoid phase errors in the side-band

spectra. Simulation parameters are summarized in Table S1 in the SI.

In the limit of fast exchange, restricted molecular motion will lead to partial averaging of anisotropic interactions and, thus, to a scaling of the observed interaction. The scaling factor, often referred to as the order parameter $S$ depends on the amplitude of the underlying motion. For the three-site exchange process considered here, it is determined by the opening angle $\theta$ (see Fig. 1a) and given by $P_2(\cos\theta)$, where $P_2$ is the second-order Legendre polynomial. In order to determine motional time scales that result in a scaling of the anisotropic interaction of interest, apparent tensor anisotropies $\delta^{\mathrm{fit}}$ were obtained by $\chi^2$ fitting. For this purpose, reference simulations without exchange were performed for a grid of interaction strengths ($\delta_{\mathrm{IS}}$ for dipolar recoupling, $\delta_{\mathrm{CSA}}$ for CSA recoupling, $\delta_{\mathrm{Q}}$ for quadrupolar simulations). All other parameters of this reference set were the same as for the simulations with exchange and the simulated time-domain data used for the fit. Since this approach neglects the signal decay due to relaxation during the recoupling (characterized by $T_2$ or $T_{1\rho}$ depending on the recoupling scheme) and relaxation effects are more significant for longer recoupling times, only the initial build-up of the recoupling curve was used for the $\chi^2$ fit in these cases. For quadrupolar and off-magic-angle spinning simulations, rapid signal decay was observed for intermediate motional time scales. In principle, the $T_2$ characterizing this line broadening is different for each crystallite and spinning sideband in the spectrum requiring a more complex data analysis to determine reliable tensor parameters. To simplify the data analysis, we approximate the effects of relaxation by assuming a single exponential decay of the total signal. Therefore, exponential line broadening was applied to the reference simulations in the time domain as $\exp(-\pi\lambda_{\mathrm{lb}}t)$ prior to $\chi^2$ fitting and a two-dimensional grid of $\delta$ and $\lambda_{\mathrm{lb}}$ used. For pulse-sequence based dipolar and CSA recoupling, including $\lambda_{\mathrm{lb}}$ in the fitting procedure only had negligible impact on the resulting $\delta^{\mathrm{fit}}$ and was, therefore, omitted (see Fig. S4 in the SI for a comparison of the fitting results). If a precise determination of the anisotropic interaction under intermediate motion is the aim, several approaches yielding better quality of the fitted tensor parameters in the intermediate motional regime have been reported in the literature. These include line shape analyses of MAS sideband spectra using analytical expressions based on Floquet theory (Schmidt et al., 1986; Schmidt and Vega, 1987) or the Anderson-Weiss formalism (Hirschinger, 2006, 2008) for cross-polarization transfer. However, these more advanced techniques are not required for the characterization of the motional time scales contributing to the partially averaged anisotropic interactions and we, therefore, opted for the simpler approach described above. Data processing was done using the Python packages numpy and matplotlib (Harris et al., 2020; Hunter, 2007) (for CSA recoupling) and Matlab (The MathWorks Inc., Natick, MA, U.S.A., all other simulations).

## 3   Results and Discussion

### 3.1   Dipolar Recoupling

Magic-angle spinning averages all second-rank anisotropic interactions and removes the heteronuclear dipolar couplings. Experimentally, there are two possible ways to reintroduce them in order to allow measurements of the dipolar order parameters. One can either use pulse sequences that interfere with the averaging by MAS, so-called dipolar recoupling sequences (Nielsen

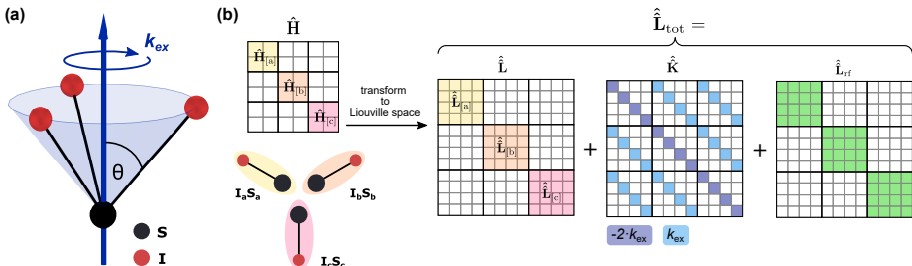

**Figure 1.** a) Schematic representation of the exchange model used to mimic restricted molecular motion. The dynamic process was modeled using a three-site jump model corresponding to a rotation around a $C_3$ symmetry axis. The time scale of this motion is characterized by the exchange-rate constant $k_{\mathrm{ex}}$ and the only difference between the three discrete states (a, b and c, shown for a heteronuclear two-spin system) is the orientation of the tensor characterizing the anisotropic interaction (dipolar coupling, CSA and quadrupolar). b) Schematic depiction of the total Liouvillian in the composite space of the three states used in simulations with dynamics. In a first step, the matrix representations of the subsystem Hamiltonians $\hat{H}_{[\mathrm{a}]}$, $\hat{H}_{[\mathrm{b}]}$ and $\hat{H}_{[\mathrm{c}]}$, containing all spin-spin and spin-field interactions relevant for the corresponding state, are computed in Hilbert space. The Liouvillian super operators for each state (e.g. $\hat{\hat{L}}_{[\mathrm{a}]}$) can then easily be computed as the commutation super operator of the subsystem Hamiltonians and combined to yield the Liouvillian $\hat{\hat{L}}$ in the composite space. Exchange between the states is included through the addition of the exchange matrix $\hat{\hat{K}}$, shown here for the symmetric three-site exchange process. Depending on the recoupling scheme, the Liouvillian computed from the radiofrequency Hamiltonian also has to be included.

et al., 2012) or one can change the angle of the sample rotation axis slightly off the magic angle (Martin et al., 2015). The first approach can be implemented using standard MAS probes while the second requires either specialized hardware to change the spinning angle during the experiment or a permanent detuning of the spinning angle leading to line broadening in all spectral dimensions. We will discuss the effects of dynamics in both implementations.

### 3.1.1 Pulsed Dipolar Recoupling

Measuring order parameters from incompletely averaged dipolar couplings under MAS usually requires the use of a pulse sequence that reintroduces the dipolar interaction. A variety of such recoupling sequences has been developed that are based on different approaches (Nielsen et al., 2012). In this work, we study the apparent recoupling behaviour of three different pulse schemes: (i) Hartmann-Hahn cross polarization (Hartmann and Hahn, 1962; Pines et al., 1972; Stejskal et al., 1977) that was proposed first and is used most often to achieve polarization transfer, (ii) Rotational-Echo Double Resonance (REDOR) (Gullion and Schaefer, 1989; Gullion, 1998) that works best in dilute spin systems under fast MAS and (iii) the more recently developed windowed Phase-Alternating R-symmetry Sequence (wPARS) (Hou et al., 2014; Lu et al., 2016) that can be used in protonated systems at moderate MAS frequencies (ca. 20 kHz) since it also performs homonuclear decoupling. Schematic depictions of the corresponding pulse schemes can be found in Fig. 2. In the CP experiment, the dipolar coupling is reintroduced by matching the rf field strengths on the two channels to one of the zero- or double-quantum Hartmann-Hahn matching conditions ($\nu_{1\mathrm{I}} \pm \nu_{1\mathrm{S}} = n\nu_{\mathrm{r}}$). In general, CP is mostly used to transfer polarization from high-$\gamma$ nuclei such as protons to low-$\gamma$ nuclei in order to increase the signal-to-noise ratio in spectra of low-$\gamma$ nuclei. However, heteronuclear dipolar couplings can be

determined by incrementing the CP contact time and measuring the full recoupling curve.

In the REDOR scheme, the dipolar coupling is reintroduced by trains of rotor synchronized $\pi$ pulses. The REDOR curve is then computed as $\Delta S(\tau)/S_0(\tau) = (S_0(\tau) - S(\tau))/S_0(\tau)$, where $S_0(\tau)$ corresponds to the signal measured in a reference experiment without the $\pi$ pulses on the I spins and $S(\tau)$ is the signal for the REDOR experiment. The normalization of the signal with respect to a reference experiment ensures that the signal decay (due to relaxation) does not need to be accounted for when fitting. The features of the curve, thus, exclusively report on the tensor parameters of the heteronuclear dipolar coupling and the detailed shape of the curve can also unambiguously reveal a non-zero tensor asymmetry that can be fit (Schanda et al., 2011a; Asami and Reif, 2019).

The wPARS experiment on the other hand uses a symmetry-based sequence (Zhao et al., 2001; Levitt, 2007) with a basic R element for the recoupling. In this recoupling scheme, a $\mathrm{R}N_0$ block and its $\pi$-phase shifted counterpart $\mathrm{R}N_\pi$ are applied on the I channel in an alternating fashion. Each of the $\mathrm{R}N$ blocks contains a standard $\mathrm{R}N_n^\nu$ cycle comprising $N$ basic R elements ($\pi$ pulses) that are synchronized with $n$ rotor cycles. Pulse phases alternate between $\phi$ and $-\phi$, where $\phi = \pi\nu/N$ is the phase shift between neighboring pairs of R elements. On the S channel, $\pi$ pulses are applied between $\mathrm{R}N_0$ and $\mathrm{R}N_\pi$ blocks in order to suppress the CSA of the I spins that would otherwise also be recoupled by the R sequence. In principle, any $\mathrm{R}N_n^\nu$ sequence that recouples the dipolar interaction can be used and we chose to simulate the $\mathrm{R}10_1^3$ sequence due to its reasonable rf requirements for moderate MAS frequencies ($\nu_1 = 5 \cdot \nu_\mathrm{r}$). Unlike REDOR, symmetry-based recoupling experiments do not inherently compensate for signal loss caused by relaxation. However, relaxation compensation can be achieved in the same way as is done in the REDOR experiment by using the symmetry-based sequence instead of the hard refocusing pulses (Chen et al., 2010). Acquiring the reference experiment without one of the central $\pi$ pulses allows for the compensation of $T_2$ signal decay during the experiment. These implementations are primarily applied in connection with quadrupolar nuclei and have not been used to measure averaged anisotropic interactions for dynamics characterization.

Examples of simulated recoupling curves for slow ($k_\mathrm{ex} = 1\ \mathrm{s}^{-1}$), intermediate ($k_\mathrm{ex} = 1 \cdot 10^3\ \mathrm{s}^{-1}$) and fast ($k_\mathrm{ex} = 1 \cdot 10^{11}$ $\mathrm{s}^{-1}$) exchange at 20 kHz MAS are shown in Fig. 2 for the three recoupling sequences. Simulations are shown for a dipolar coupling with $\delta_\mathrm{IS}/(2\pi) = 5$ kHz and a motional amplitude of $\theta = 70.5°$ (further examples can be found in Figs. S6, S7 and S8 in the SI). The recoupling curves for all three sequences show characteristic oscillations. The frequency of these oscillations depends on the residual dipolar coupling and, thus, on the scaling factor that results for the specific pulse scheme. As expected, the oscillation frequency is reduced for fast exchange due to the scaling of the anisotropic interaction by the rapid molecular motion. In the intermediate exchange regime, strong damping of the oscillation is observed. Additionally, a decay of magnetization is observed for CP when strong dipolar couplings and longer contact times are considered (see Fig. S1 in the SI).

The apparent $\delta_\mathrm{IS}^\mathrm{fit}$ can be extracted from the simulated recoupling curves by comparison with a set of reference simulations without exchange. Due to the damping of the oscillations in the intermediate exchange regime (see Fig. 2), only the initial

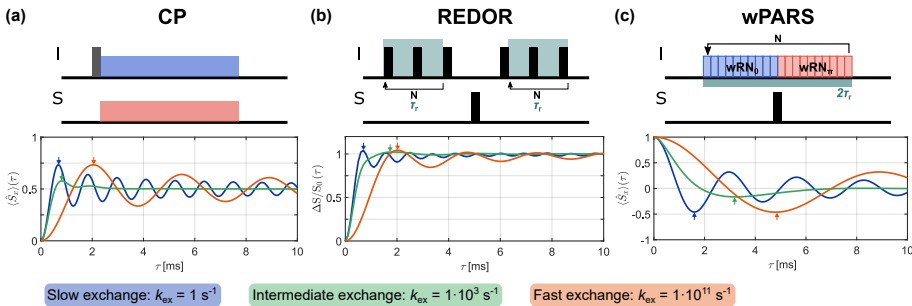

**Figure 2.** Examples of simulated dipolar recoupling curves for (a) CP, (b) REDOR, and (c) wPARS for slow, intermediate and fast exchange (20 kHz MAS, $\delta_{IS}/(2\pi) = 5$ kHz, $\theta = 70.5°$). In the intermediate exchange regime, a damping of the oscillations is observed while fast exchange simply leads to a scaling of the dipolar coupling and, thus, a reduction of the oscillation frequency. The apparent $\delta_{IS}$ is determined by $\chi^2$-fitting a set of reference simulations with different heteronuclear dipolar couplings without exchange to the initial build-up of the recoupling curve (up to the first local extremum, position indicated by arrows).

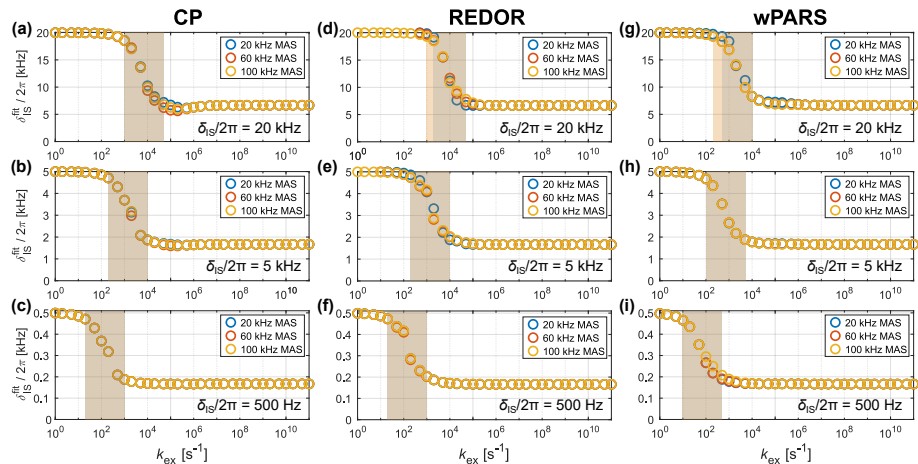

**Figure 3.** Fitted apparent anisotropy of the dipolar-coupling tensor $\delta_{IS}^{fit}$ for CP (a-c), REDOR (d-f) and wPARS (g-i) for different MAS frequencies (20, 60 and 100 kHz) and dipolar-coupling strengths ($\delta_{IS}/(2\pi) = 20$, 5 and 0.5 kHz). The underlying molecular motion was modeled using a three-site jump process with an opening angle of $\theta = 70.5°$ (order parameter $S_{IS} = -1/3$), which corresponds to the case of a $sp^3$-hybridised carbon site. The rf field strengths used for the recoupling are summarized in Table S2 in the SI. For slow exchange, the full dipolar coupling is retained, while the scaled interaction is observed for fast exchange. In the intermediate exchange regime, a smooth transition from the full to the scaled coupling is observed (shaded area, defined as the region where the difference between two consecutive fitted $\delta_{IS}^{fit}$ exceeds 6 % of the difference between the full and scaled $\delta_{IS}$ used in the simulations). The position of this transition region depends on the coupling strength and occurs at slower motions for smaller couplings. The MAS frequency only has a marginal effect on the location of the intermediate regime.

build-up of the curve up to the first local extremum was used for the $\chi^2$-fit. In principle, the observed decay of the recoupling curve in the intermediate regime can be included in the fit by using a two-dimensional grid with an additional line-broadening parameter $\lambda_{\text{lb}}$. However, no change of the obtained $\delta_{\text{IS}}^{\text{fit}}$ ensued for such a 2D grid (see Fig. S4 in the SI for a comparison of the two fitting routines) and the results presented here stem from fits without $\lambda_{\text{lb}}$. The resulting $\delta_{\text{IS}}^{\text{fit}}$ as a function of the exchange-rate constant $k_{\text{ex}}$ are shown in Fig. 3 for different MAS frequencies and dipolar-coupling strengths. For slow exchange, the full (unscaled) dipolar coupling is observed while fast exchange results in the scaling of the anisotropic interaction by a factor of $1/3$ (as expected for an opening angle of $\theta = 70.5°$). A smooth transition from the full to the scaled interaction is observed in the intermediate exchange regime. The transition region is shaded to facilitate the visual comparison between different sets of simulations and is defined as the region where the difference between two consecutive fitted $\delta_{\text{IS}}^{\text{fit}}$ exceeds $6\%$ of the difference between the full and scaled $\delta_{\text{IS}}$ used in the simulations. The position of this transition region depends strongly on the strength of the interaction. For weaker dipolar couplings, slower motion results in the scaling of the observed coupling and the transition region occurs for smaller values of $k_{\text{ex}}$. The transition region roughly spans motional time scales over two orders of magnitude between $1/10$ and $10$ times the magnitude of the dipolar coupling. However, the exact position varies slightly depending on the recoupling sequence used. For all three pulse sequences investigated, only a negligible dependence on the MAS frequency is observed. Simulations at 500 kHz MAS (see Fig. S3 in the SI) further suggest that the influence of the MAS frequency will remain unimportant even if significant advances in the achievable spinning frequency are realized in the future. Similar results are obtained for other CP matching conditions (different rf fields at the same MAS frequency, see Fig. S2 in the SI) and REDOR simulations with different $\pi$ pulse lengths (see Fig. S5 in the SI). The rf field strength, therefore, does not seem to influence the transition from the full to the scaled coupling significantly.

In the limit of fast exchange, the order parameter of the dynamic process and, thus, the opening angle $\theta$ in our three-site jump model (see Fig. 1a for the bond geometry) determines the scaling of the motion. Figure 4a-c shows a comparison of the fitted apparent $\delta_{\text{IS}}^{\text{fit}}$ for different opening angles for a dipolar coupling of $\delta_{\text{IS}}/(2\pi) = 5$ kHz at a MAS frequency of 20 kHz for CP, REDOR and wPARS. As expected, more restricted motion leads to a larger scaling factor for the incompletely averaged coupling. However, the position and width of the transition region does not seem to be affected significantly.

All three recoupling sequences presented here can be modified to allow the scaling of the effective dipolar coupling. Based on the observed dependence of the position of the transition region on the strength of the anisotropic interaction (see Fig. 3), one could expect that this will enable studying different motional time scales. In the CP experiment, the dipolar coupling strength can be scaled down by tilting one of the two applied spinlock fields away from the transverse plane (van Rossum et al., 2000; Hong et al., 2002). This is schematically depicted in Fig. 4d. The extent of the scaling is characterized by the tilt angle $\vartheta_1$ (where $\vartheta_1 = 90°$ corresponds to the unscaled "normal" CP experiment). Figure 4d shows a comparison of the resulting $\delta_{\text{IS}}^{\text{fit}}$ for $\vartheta_1 = 90°$ and $20°$ for a dipolar coupling anisotropy of 5 kHz. Examples of recoupling curves for different tilt angles are shown in Fig. S6 in the SI. Changing the tilt angle of the applied rf field on one of the channels does indeed affect the intermediate exchange regime where the transition from the full to the motionally averaged dipolar coupling occurs. The magnitude of the

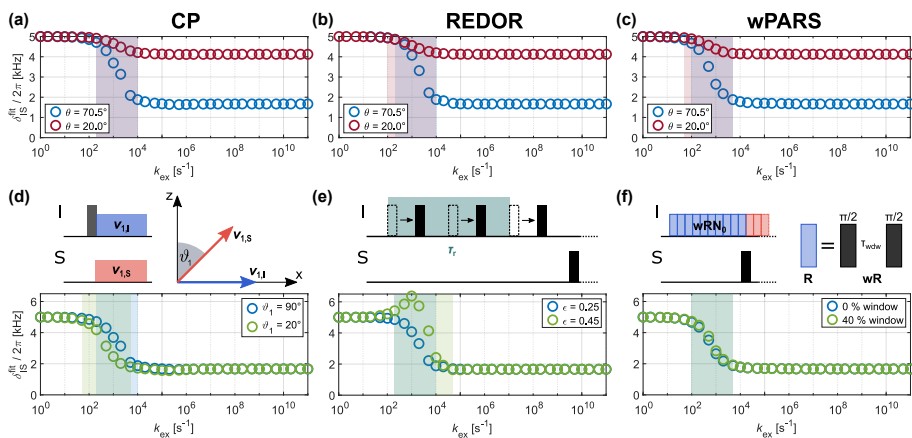

**Figure 4.** Fitted apparent $\delta_{\mathrm{IS}}^{\mathrm{fit}}$ for CP, REDOR and wPARS for: a-c) different opening angles $\theta$ (see Fig. 1a for the geometry of the jump process) and d-f) modifications of the pulse sequence that lead to a scaling of the effective heteronuclear dipolar coupling. Data is shown for a MAS frequency of 20 kHz and $\delta_{\mathrm{IS}}/(2\pi) = 5\,\mathrm{kHz}$. Examples of the corresponding recoupling curves can be found in Figs. S6, S7 and S8 in the SI. In contrast to the amplitude of the motion, the modification of the pulse sequence can affect the transition region (shaded area, defined as the region where the difference between two consecutive fitted $\delta_{\mathrm{IS}}^{\mathrm{fit}}$ exceeds 6 % of the difference between the full and scaled $\delta_{\mathrm{IS}}$ used in the simulations).

effective dipolar coupling scales with $\cos\vartheta_1$ requiring exceedingly small angles $\vartheta_1$ to achieve a significant scaling. Therefore, the magnitude of the experimentally achievable shift in the transition region will be limited. We do not expect that significant gain in information on the distribution of motions can be obtained from such angle-dependent measurements due to the inher-
ent low precision of the determined order parameters.

For the REDOR experiment, several schemes exist that allow scaling down the effective dipolar coupling. Strong dipolar couplings result in a rapid build-up of the REDOR curve, which often limits the amount of points on the curve that can be measured experimentally before the signal has decayed. In these cases, REDOR schemes that involve shifting the position of
the rotor synchronized $\pi$ pulses are often used. Here, we study the two-pulse shifted REDOR scheme (Jain et al., 2019a), in which the position of both $\pi$ pulses within a rotor period is altered while keeping the time separation between them constant at $0.5\tau_{\mathrm{r}}$ (see Fig. 4e). This results in the scaling of the effective dipolar coupling by $\sin(2\pi\epsilon)$, where $\epsilon$ characterizes the pulse shift (see Fig. S7 in the SI for further details) and the classic REDOR experiment corresponds to $\epsilon = 0.25$. Apparent $\delta_{\mathrm{IS}}^{\mathrm{fit}}$ for $\epsilon = 0.25$ and 0.45 are shown in Fig. 4e for $\delta_{\mathrm{IS}}/(2\pi) = 5\,\mathrm{kHz}$. In the limit of fast and slow exchange, the simulated REDOR
curves for $\epsilon = 0.45$ and $\delta_{\mathrm{IS}}/(2\pi) = 5\,\mathrm{kHz}$ (corresponding to $\delta_{\mathrm{IS}}^{\mathrm{eff}}/(2\pi) = \sin(2\pi\epsilon)\cdot\delta_{\mathrm{IS}}/(2\pi) = 1545\,\mathrm{Hz}$) agree well with those obtained for a dipolar coupling with $\delta_{\mathrm{IS}}/(2\pi) = 1545\,\mathrm{Hz}$ in a classic REDOR experiment (see Fig. S7 in the SI). This indicates that shifting the pulse positions in these exchange regimes (ca. $k_{\mathrm{ex}} < 1\cdot10^2\,\mathrm{s}^{-1}$ and $k_{\mathrm{ex}} > 1\cdot10^5\,\mathrm{s}^{-1}$ for this particular coupling strength) has the desired scaling effect. However, changing the position of the $\pi$ pulses strongly affects the appearance of the REDOR curve in the intermediate exchange regime. A shift parameter of $\epsilon \neq 0.25$ leads to a rapid build-up of the

REDOR curve and removes the characteristic oscillations. Extracting the apparent $\delta_{\text{IS}}^{\text{fit}}$ for motion on these time scales $(1 \cdot 10^2$ $\text{s}^{-1} < k_{\text{ex}} < 1 \cdot 10^5 \text{ s}^{-1}$ for $\delta_{\text{IS}}/(2\pi) = 5$ kHz) is, therefore, impractical and the resulting values contain no real information (see Fig. 4e for fit results). This pulse-shifted implementation of the REDOR experiment thus seems to only be suitable for sufficiently slow or fast dynamics where the full or the motionally averaged interaction is observed. The time scales of these "fast" or "slow" motions depend on the strength of the unscaled dipolar coupling itself (see Fig. 3g-i).


The wPARS sequence allows scaling up the effective dipolar coupling by introducing a window without rf irradiation in the basic R element (see Fig. 4f for a schematic depiction) (Lu et al., 2016). The larger the fraction of time of this window in the basic element, the larger the observed effective coupling. Examples of simulated recoupling curves for different window lengths are shown in Fig. S8 in the SI for a dipolar coupling anisotropy of 5 kHz. Similar to our observations for CP, the effect
of the dynamic process on the appearance of the recoupling curves is the same for all window lengths. However, the range of scaling factors that can be achieved is too small to result in a considerable shift of the transition region from the full to the motionally averaged coupling (see Fig. 4f).

In the intermediate exchange regime, line broadening due to the underlying dynamics and thus damped oscillations are ob-
served during the recoupling (see Fig. 2). This signal decay is due to transverse relaxation and depends on the coupling strength and the experimental parameters (MAS frequency, pulse sequence, etc.). In the CP experiment, the signal loss is characterized by $T_{1\rho}$ due to the applied spin lock and mainly affects the recoupling for long contact times (see Fig. S1 in the SI). Since the width of the dipolar coupling tensor is encoded in the initial build up of the recoupling curve occurring on a time scale that is considerably faster than the $T_{1\rho}$ decay, the apparent tensor anisotropy can still be extracted in most cases. In the REDOR
experiment, the signal loss due to transverse relaxation (characterized by $T_2$) is compensated by normalizing the dephasing signal with the $S_0$ curve obtained in the reference experiment. Nevertheless, motion on an intermediate time scale strongly attenuates the oscillations in the REDOR curve which has detrimental effects on the quality of the fitted tensor parameters. In the wPARS experiment, the line broadening during the recoupling is also characterized by $T_2$ and leads to damping of the oscillations in the dephasing curve. However, the sequence has no inherent compensation of the signal loss making it susceptible
to relaxation effects. Although a suitable reference experiment can be designed (Chen et al., 2010), no such experiments have been used for the characterization of dynamic time scales based on scaled anisotropic interactions to the best of our knowledge.

In our simulated data we were able to circumvent relaxation-related issues by only fitting the initial build up of the recoupling curves for all three sequences. Since the simulated curves are ideal and noise-free, the initial slope characterizes the magnitude
of the effective coupling perfectly resulting in smooth transitions of the fitted tensor anisotropy. Fitting longer mixing times produced strongly varying results for the magnitude of the scaled coupling in the transition region. In experimental spectra, however, it is generally advisable to fit the observed oscillations in order to get an unambiguous result for the effective coupling strength. Moreover, it is important to confirm that the dynamic averaging takes place within the fast motion limit to ensure that the measured order parameter is well-defined. Although the observation of overdamped oscillations (see Fig. 2) indicates

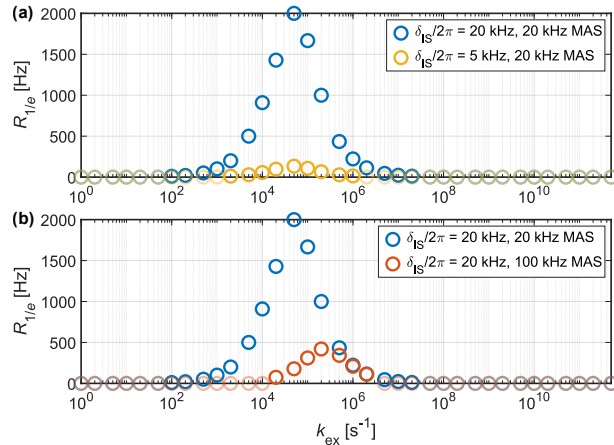

**Figure 5.** Characterization of the observed transverse relaxation during the REDOR reference experiment ($S_0$) for different dipolar coupling strengths (a) and MAS frequencies (b). Shown is the extracted $R_{1/e} = \frac{1}{T_{1/e}}$ relaxation rate as a function of the exchange rate constant, where $T_{1/e}$ corresponds to the time required for the signal to decay below a value of $1/e$. Data points shown with reduced opacity correspond to cases where the $T_{1/e}$ time exceeds the observation window of 20 ms. An opening angle of $\theta = 70.5°$ was assumed for the three-site exchange process. The simulated $S_0$ curves are shown in Fig. S12 in the SI. Rapid signal decay (large $R_{1/e}$) is observed for a broader range of motional time scales for stronger couplings and slower MAS. Moreover, faster MAS shifts the region where strong relaxation is observed to faster motion.

motion on an intermediate time scale, additional measurements should be performed to determine the dynamic regime. Such insights can for example be obtained by characterizing $T_2$ signal losses during the recoupling. In the case of REDOR, this information can be extracted from the signal loss in the $S_0$ curves obtained in the reference experiment. Figure S12 in the SI shows examples of simulated $S_0$ curves as a function of the exchange-rate constant for different dipolar coupling strengths and MAS frequencies. The signal loss in this echo reference experiment is determined by $T_2'$. However, since the decay is not mono-exponential in solids, it is often characterized qualitatively by the $R_{1/e} = \frac{1}{T_{1/e}}$ relaxation rate (see Fig. 5), where $T_{1/e}$

corresponds to the time required for the signal to decay below a value of $1/e$. Short decay times are diagnostic for the presence of motion on an intermediate time scale. While the magnitude of the relaxation rate mainly depends on the coupling strength, the exact position of the $T_2$ minimum depends on the MAS frequency and can also be predicted using Redfield relaxation theory (Schanda and Ernst, 2016). For fast MAS (around 100 kHz), large relaxation rates are observed for motion on time scales

corresponding to the beginning of the fast motion regime, while the region with rapid signal decay extends into the transition region for slow MAS (20 kHz).

In addition to the effects during the recoupling period, line broadening will also affect other time periods in the experiments. Decay of transverse magnetization ($T_2$ relaxation) during the detection period for example will broaden spectral lines and

reduce resolution and sensitivity. The observed $T_2$ will depend on a variety of factors, e.g. the coupling strength, the MAS frequency and the decoupling scheme employed (Schanda and Ernst, 2016). Moreover, dynamics on the intermediate time scale

have been shown to have detrimental effects on polarization-transfer experiments (Nowacka et al., 2013; Aebischer and Ernst, 2024) and will, thus, reduce the signal-to-noise ratio even leading to entire molecular segments missing in spectra (Callon et al., 2022). A detailed discussion of these effects is beyond the scope of the manuscript.


### 3.1.2 Off-Magic-Angle Spinning

Instead of using rf irradiation to reintroduce ansiotropic interactions under MAS, off-magic-angle spinning (off-MAS) can be used to measure order parameters. Changing the tilt of the sample spinning axis with respect to the external $B_0$ field away from the magic angle reintroduces a scaled anisotropic interaction. The magnitude of the scaled interaction depends on the
offset from the magic angle $\theta_{\mathrm{rot}} = \theta_{\mathrm{m}} + \Delta$, where $\theta_{\mathrm{rot}}$ corresponds to the angle between the external field and the rotation axis and the scaling factor is given by $P_2(\cos\theta_{\mathrm{rot}})$. Experimentally, the reintroduction of the scaled interaction will result in scaled powder patterns and information on any underlying motion can be gained from a line shape analysis. This was first used to study molecular re-orientation by fitting CSA line shapes in one- and two-dimensional $^{13}$C CPMAS spectra for different offset angles (Schmidt and Vega, 1989; Blümich and Hagemeyer, 1989) and has also been extended to quadrupolar nuclei
(Kustanovich et al., 1991). However, large angle offsets significantly deteriorate spectral resolution. For heteronuclear dipolar couplings, residual couplings can also manifest in perturbations of the $J$ modulation observed in a spin-echo experiment. In this case, small offset angles $|\Delta| < 0.5°$ suffice to introduce significant residual couplings in directly bound spin pairs without notably deteriorating the spectral resolution. Such measurements were first demonstrated for homonuclear dipolar couplings in $^{13}$C spin pairs (Pileio et al., 2007) but have since also been used to determine order parameters in backbone amides (Xue et al.,
2019b) and methyl groups (Xue et al., 2019a) in deuterated protein samples.

Following the work of Pileio et al. (2007), the powder-averaged dephasing signal for a scalar coupled heteronuclear spin pair under off-MAS with small angle offsets is approximately given by

$$S_{\mathrm{mod}}(\tau, \Delta) \approx \frac{1}{2} \int\limits_0^\pi \cos\left(\pi J\tau - \sqrt{2}\Delta\frac{\delta_{\mathrm{IS}}}{2}P_2(\cos\beta_{\mathrm{PR}})\tau\right)\sin(\beta_{\mathrm{PR}})\,d\beta_{\mathrm{PR}}\,, \tag{1}$$

where $\delta_{\mathrm{IS}}$ corresponds to the anisotropy of the dipolar coupling, $\beta_{\mathrm{PR}}$ denotes the angle between the internuclear vector and the rotor axis and $P_2$ is the second-order Legendre polynomial. The offset of the rotation angle from the magic angle is given by $\Delta$ and can be positive or negative. Depending on the relative sign of the scalar $J$ and the dipolar coupling, positive or negative angle offsets can reduce or increase the observed modulation frequency.

Figure 6a-c shows examples of simulated dephasing curves in a heteronuclear spin pair with parameters based on a backbone amide group (NH, $J = $ -90 Hz, $\delta_{\mathrm{IS}}/(2\pi) = 21$ kHz, corresponding to an effective N-H distance of 1.05 Å) for slow ($k_{\mathrm{ex}} = 1 \cdot 10^{-2}$ s$^{-1}$), intermediate ($k_{\mathrm{ex}} = 1 \cdot 10^3$ s$^{-1}$) and fast exchange ($k_{\mathrm{ex}} = 1 \cdot 10^{11}$ s$^{-1}$) and different angle offsets. In principle,

positive offset angles should increase the observed modulation frequency due to the opposite signs of the dipolar and the $J$ coupling. This is indeed observed in the case of slow exchange. However, the underlying three-site jump process for an opening angle of $\theta = 70.5°$ (see Fig. 1a) results in an order parameter of $-1/3$ and thus a sign change for the apparent $\delta_{IS}$. Smaller opening angles ($\theta < 54.74°$) of the jump model will lead to a positive order parameter and no sign change in the value of the residual dipolar coupling. Positive angle offsets, therefore, reduce the oscillation frequency in the limit of fast exchange for our set of parameters. Larger offset angles result in a more significant distortion of the modulated signal. In the intermediate exchange regime, relaxation results in a decay of magnetization leading to a damping of the oscillation.

As described in the Methods section, the apparent $\delta_{IS}$ was determined by $\chi^2$-fitting a set of reference simulations without exchange (ca. 80 ms observation window). In order to account for the observed signal decay due to relaxation, a two-dimensional grid with the additional line broadening parameter $\lambda_{lb}$ was used for the fitting. The resulting $\delta_{IS}^{fit}$ and the corresponding $\lambda_{lb}$ are shown in Fig. 6d for different spinning frequencies ($\Delta = 0.05°$) and Fig. 6e for different angle offsets (20 kHz spinning frequency). In the limit of slow ($k_{ex} < 1 \text{ s}^{-1}$) and fast exchange ($k_{ex} > 1 \cdot 10^7 \text{ s}^{-1}$), the full and scaled dipolar coupling are obtained as expected. For intermediate exchange ($k_{ex} \approx 1 \cdot 10^5 \text{ s}^{-1}$) the oscillations in the dephasing curves are damped and no meaningful information on $\delta_{IS}$ can be gained (see Fig. S9 in the SI for contour plots of $\chi^2$). The observed line broadening strongly depends on the spinning frequency and is reduced for faster spinning (see Fig. 6d).

Compared to simulations of pulsed dipolar recoupling under MAS for the same coupling strength (see Fig. 3) the transition from the full to the incompletely averaged interaction occurs for significantly slower motion. This can be attributed to the scaling of the dipolar coupling by the small angle offset (see Eq. (1)) and the position of the transition can be shifted by changing the angle offset (see Fig. 6e). The scaling of the dipolar coupling by off-MAS does not affect the motional time scales for which rapid relaxation is observed. Therefore, the range of exchange-rate constants where the transition towards the scaled interaction occurs is separated from the regions where the signal decays fast. This separation is further improved for even faster spinning frequencies (see Fig. S10 in the SI for simulation results at 500 kHz spinning), suggesting off-MAS as a suitable method for characterizing dynamics at fast MAS.

For exchange-rate constants around $1 \cdot 10^2 \text{ s}^{-1}$, the sign of the anisotropy of the dipolar coupling is not well defined (see Fig. S9 in the SI for more details), leading to jumps in the resulting $\delta_{IS}^{fit}$ (see Fig. 6d and e). In this exchange regime, the dipolar coupling is scaled to values close to zero since the underlying three-site jump leads to a sign change of $\delta_{IS}$ for faster motion. The jumps in the fitted $\delta_{IS}^{fit}$ can thus be attributed to the sign change of the dipolar coupling for faster motion. For a lower amplitude of motion (see Fig. 6f for simulation results for an opening angle of $\theta = 20.0°$ corresponding to an order parameter of $S_{IS} \approx 0.82$) no such jumps in the fitted $\delta_{IS}^{fit}$ are observed. In this case, only the line broadening in the intermediate exchange regime deteriorates the fit quality.

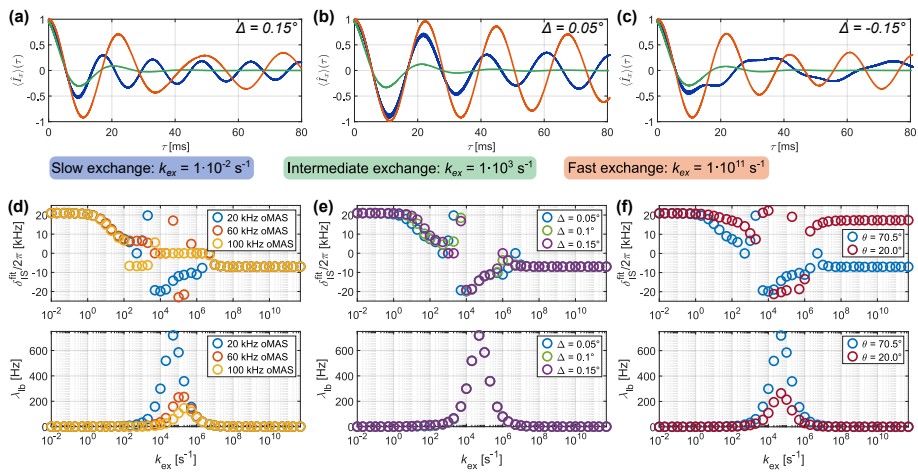

**Figure 6.** Simulated off-magic angle spinning of a heteronuclear NH two-spin system with a scalar coupling of $J = -90$ Hz and a dipolar coupling of $\delta_{IS}/(2\pi) = 21$ kHz. a-c) Examples of simulated dephasing of proton transverse magnetization $\hat{I}_x$ for spinning at different offset angles $\Delta$ (20 kHz spinning frequency, $\theta = 70.5°$). d-f) Fitted $\delta_{IS}^{fit}$ and line broadening parameter $\lambda_{lb}$ as a function of the exchange-rate constant for off-angle spinning simulations: d) $\Delta = 0.05°$ and different spinning frequnecies, e) 20 kHz spinning and different angle offsets, f) $\Delta = 0.05°$ and 20 kHz spinning for different motional amplitudes (see Fig. 1a for the underlying exchange process).

## 3.2 CSA Recoupling

The chemical-shift anisotropy, like the dipolar coupling, is averaged by molecular motion. We performed CSA simulations using a symmetry-based sequence ($R18_1^7$); this class of $RN_n^\nu$ sequences are among the most popular techniques for CSA re-
coupling (Levitt, 2007; Hou et al., 2012). The sequences consists of a train of $\pi$ pulses with alternating phases $\pm\phi = \pm\pi\nu/N = \pm70° \cdot (\pi/180°)$, applied to the nucleus of which the CSA is to be recoupled. The rf-field strength is chosen such that $N$ (here: $N = 18$) $\pi$ pulses fit into $n$ (here: $n = 1$) rotor periods; in the case of $R18_1^7$, the nutation frequency of the rf field is, thus, nine times the MAS frequency. The CSA parameters can be obtained from the evolution of the signal amplitude as a function of the duration of the recoupling sequence, by either fitting the time-domain or the frequency-domain signal. Here, we fitted the
apparent CSA tensor anisotropy, $\delta_{CSA}^{fit}$, in the time domain with a $\chi^2$ minimization procedure, comparing the simulations with dynamics against a grid of simulated rigid-limit recoupling trajectories. As for the dipolar recoupling, only the initial build-up of the curves was fitted to reduce effects of signal loss due to relaxation.

Figure 7a shows examples of CSA recoupling trajectories for slow ($k_{ex} = 1$ s$^{-1}$), intermediate ($k_{ex} = 1 \cdot 10^3$ s$^{-1}$) and fast
exchange ($k_{ex} = 1 \cdot 10^{11}$ s$^{-1}$). As for the dipolar recoupling (see Fig. 2), the frequency of the modulation is high, and identical to the rigid case, for slow exchange, and scaled down for very fast exchange. In the intermediate regime, the recoupling trajectory shows strong dampening and decays to zero. The fitted apparent $\delta_{CSA}^{fit}$ is plotted against the time scale of the underlying motion in Fig. 7b-d for different CSA strengths of $\delta_{CSA}/(2\pi) = 20, 5$ and $0.5$ kHz, respectively. The transition from the slow regime,

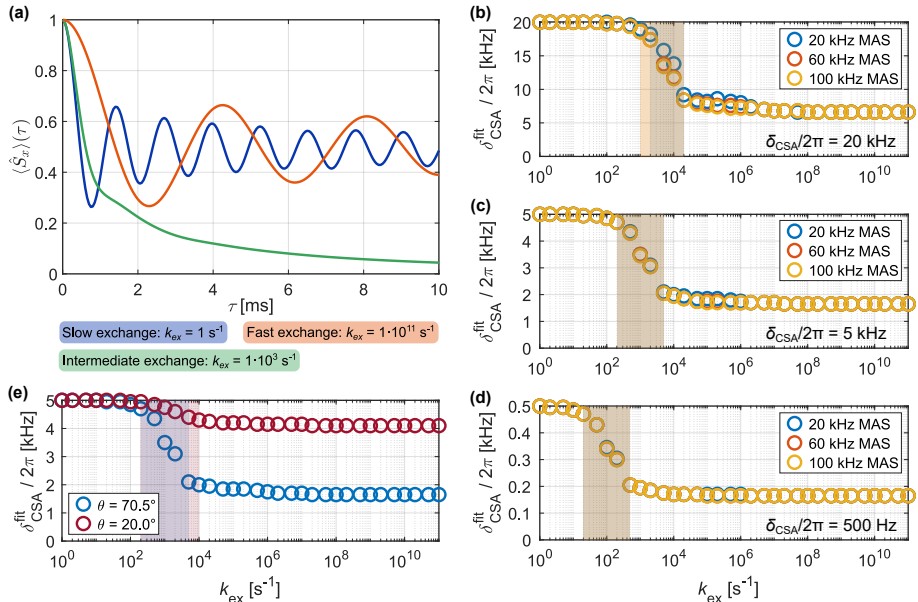

**Figure 7.** Apparent recoupling behaviour for $\mathrm{R}18_1^7$ recoupling of the CSA interaction in the presence of molecular motion. a) Examples of recoupling curves for slow, intermediate and fast exchange (20 kHz MAS, $\delta_{\mathrm{CSA}}/(2\pi) = 5$ kHz, $\theta = 70.5°$). For fast exchange, a lower frequency oscillation corresponding to the scaled CSA interaction is observed. Motion on an intermediate time scale results in a strong damping of the oscillations and signal decay. b-d) MAS dependence of fit results for different interaction strengths. e) Comparison of fitted apparent $\delta_{\mathrm{CSA}}^{\mathrm{fit}}$ for different opening angles $\theta$ (20 kHz MAS, $\delta_{\mathrm{CSA}}/(2\pi) = 5$ kHz).

where the CSA is not averaged, to the fast regime is found to depend on the rigid-limit tensor anisotropy. The larger the rigid-limit CSA tensor, the shorter the time scale at which the transition from the fast regime to the slow regime occurs. For example, the averaged interaction is observed for an exchange-rate constant exceeding $k_{\mathrm{ex}} = 5 \cdot 10^3 \ \mathrm{s}^{-1}$ if the rigid-limit chemical-shift anisotropy is 20 kHz, whereas the transition is found at approximately $k_{\mathrm{ex}} = 1 \cdot 10^2 \ \mathrm{s}^{-1}$ if the anisotropy is 0.5 kHz. The opening angle $\theta$ of the underlying jump model (see Fig. 1a) also changes the scaling of the CSA at fast time scales (Fig. 2e) but has no significant effect on the width and position of the transition region. Overall, the CSA recoupling shows similar trends as the dipolar recoupling in terms of the time scales over which it reports averaging.

### 3.3 Quadrupoles

In addition to dipolar couplings and CSA tensors, incompletely averaged quadrupolar couplings can be used to study dynamics in solid-state NMR (Shi and Rienstra, 2016; Akbey, 2022, 2023). Under MAS, the first-order quadrupolar coupling becomes time-dependent and results in spinning sideband patterns while the second-order quadrupolar coupling leads to line broadening and isotropic shifts. The intensity distribution of these sideband spectra can be used to determine the anisotropy of the quadrupolar coupling and, thus, reveals information on the time scale and amplitude of the motional process. In biological systems, deuterium ($^2$H) is often used in such studies, where it is introduced either uniformly or selectively to replace a specific

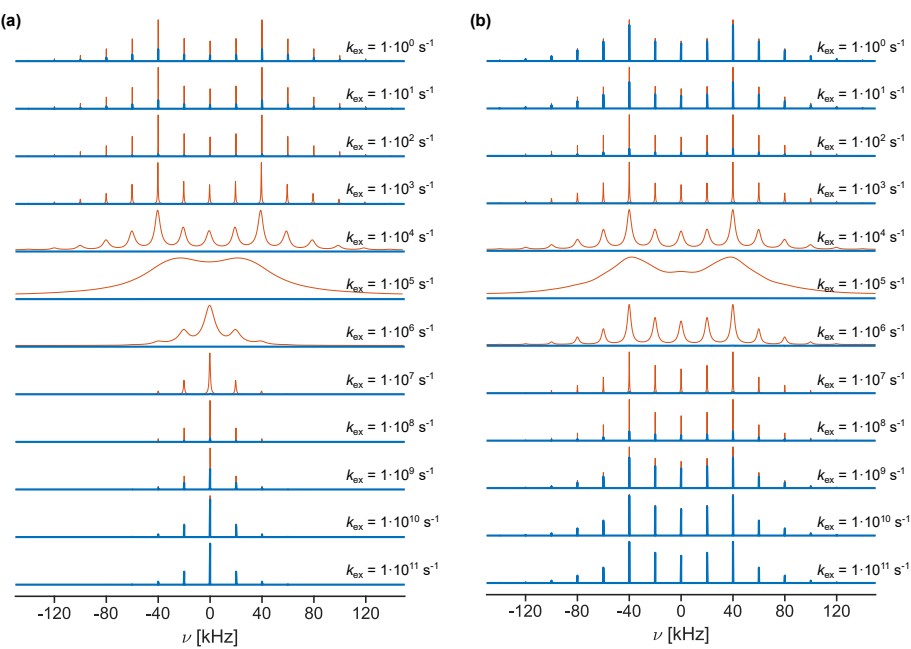

**Figure 8.** Simulated spectra of $^2$H with $\delta_Q/(2\pi) = 80$ kHz at 20 kHz MAS for different exchange-rate constants for opening angles of $\theta = 70.5°$ (a) and $20°$ (b) (see Fig. 1a for the geometry of the exchange process). The intensity of spectra shown as thick blue lines was normalized to the maximum intensity observed for all exchange-rate constants while spectra shown as thin red lines were normalized to their respective maximum intensity. All spectra were processed with 10 Hz exponential line broadening. For exchange-rate constants around $1 \cdot 10^5$ s$^{-1}$, the sideband manifold is broadened beyond detection due to rapid relaxation.

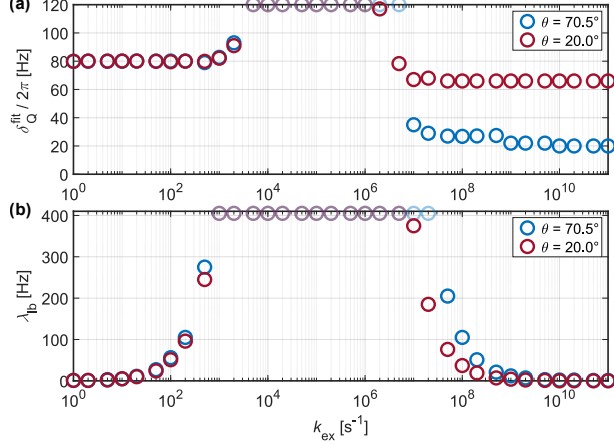

**Figure 9.** Fitted $\delta_Q^{\text{fit}}$ (a) and line broadening parameter $\lambda_{\text{lb}}$ (b) for simulations of $^2$H with $\delta_Q/(2\pi) = 80$ kHz at 20 kHz MAS for different opening angles of the three-site jump model (see Fig. 1a for the geometry of the exchange process). In the intermediate-exchange regime, the observed line broadening exceeds the reference grid of $\delta_Q$ and $\lambda_{\text{lb}}$ used for the $\chi^2$-fit, resulting in a plateau of fitted values (data points shown with reduced opacity).

$^1$H nucleus. Deuterium has a spin-1 with a quadrupolar coupling constant $C_{\text{qcc}}$ of approximately 160 kHz. Although used less often than dipolar couplings or CSA tensors, the $^2$H line shapes in static samples or the spinning sideband pattern under MAS are commonly analyzed to probe protein dynamics (Hologne et al., 2006; Shi and Rienstra, 2016; Vugmeyster and Ostrovsky, 2017; Akbey, 2023).

Figure 8 shows examples of simulated side-band manifolds for deuterium (assuming $\delta_{\text{Q}}/(2\pi) = 80$ kHz or $C_{\text{qcc}} = 160$ kHz) undergoing a symmetric three-site exchange process at 20 kHz MAS. The quadrupolar tensors of the three sites were assumed to be axially symmetric and aligned with the bond geometry depicted in Fig. 1a. As expected, fast exchange results in the scaling of the quadrupolar coupling and, thus, a narrower side-band spectrum is observed. The extent of the scaling is again dependent on the opening angle of the three-site jump process and more restricted motion leads to a scaling factor closer to one. In the intermediate exchange regime (roughly $1 \cdot 10^3 \text{ s}^{-1} < k_{\text{ex}} < 1 \cdot 10^7 \text{ s}^{-1}$) strong line broadening is observed. As described in the Methods Section, apparent $\delta_{\text{Q}}^{\text{fit}}$ were obtained by $\chi^2$ fitting a two-dimensional grid of reference simulations without exchange and an additional line broadening parameter $\lambda_{\text{lb}}$. An observation window of approximately 150 ms corresponding to a signal intensity of less than 1 % for a spectral line with 10 Hz full width at half maximum was used in the fitting procedure. Prior to $\chi^2$ fitting, a frequency shift (implemented as a first-order phase correction in time domain) was applied in time-domain to ensure that the central peak in the sideband manifold has a frequency offset of zero. Fit results are shown in Fig. 9 for two different opening angles of the jump process. Compared to the dipolar and CSA interaction, significantly faster motion ($k_{\text{ex}} > 1 \cdot 10^8 \text{ s}^{-1}$) is required to average the quadrupolar coupling due to its large magnitude. Moreover, the transition region from the full to the motionally averaged interaction is broader than for the CSA and the dipolar coupling. In the intermediate exchange regime, strong line broadening leads to featureless spectra and the true $\lambda_{\text{lb}}$ of the simulated FIDs exceeds the values considered in the grid for the fit. As discussed before, each of the side bands has a different line width (Suwelack et al., 1980; Long et al., 1994) which has to be taken into account if exact coupling parameters are required. In the limit of fast exchange, the quality of the $\chi^2$ fit deteriorates (see Fig. S11 in the SI) due to slight differences in the frequency offset of the central peak in the sideband manifold. Nevertheless, the expected scaling depending on the order parameter of the dynamic process is observed. This suggests that the measurement of the quadrupolar coupling can only give insight in the limit of fast or sufficiently slow motion and agrees with previous investigations of the effects of motion on intermediate time scales on quadrupolar side-band spectra (Kristensen et al., 1992).

### 3.4 Multiple Motions

Molecular motion is usually more complex than a simple rotation about an axis and often several motions on different time scales occur simultaneously. In order to study potential effects in such systems, we extended the three-site exchange model to a nine-site jump process that encompasses two independent rotations about non-collinear $C_3$-axes (see Fig. 10a). The inner motion is modeled by a three-site jump process with an amplitude described by $\theta^{(1)}$ within subsets of three sites. The jump process describing the outer motion leads to exchange between sites within the different subsets. Its amplitude is defined by

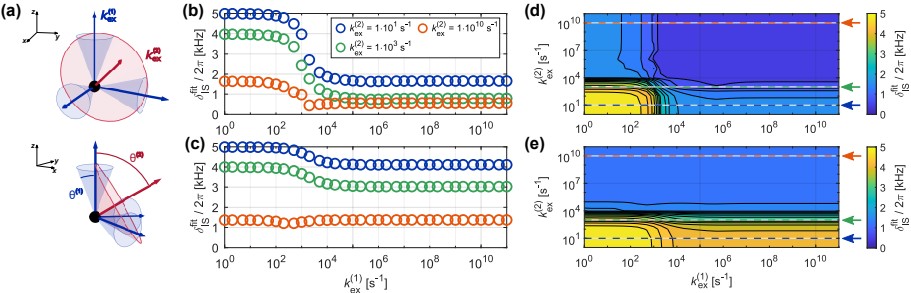

**Figure 10.** a) Schematic representation of the exchange process used to model more complex motion as a simultaneous inner and outer motion. The inner motion is described by a three-site jump process modeling rotation about the inner $C_3$-axis (blue arrows, rotation on blue cones with an opening angle of $\theta^{(1)}$), while the outer motion rotates the subsets of three sites each about the outer $C_3$-axis (red arrow, rotation on red cone with an opening angle of $\theta^{(2)}$). b-e) Apparent recoupling behaviour for a dipolar coupled heteronuclear two-spin system ($\delta_{IS}/(2\pi) = 5$ kHz) for CP recoupling at 20 kHz MAS ($\nu_{1I} = 93$ kHz, $\nu_{1S} = 73$ kHz). b-c) Fitted $\delta_{IS}^{fit}$ as a function of $k_{ex}^{(1)}$ for different $k_{ex}^{(2)}$ for $\theta^{(1)} = \theta^{(2)} = 70.5°$ (b) and $\theta^{(1)} = 20.0°$ and $\theta^{(2)} = 70.5°$ (c). d-e) Contour plots of $\delta_{IS}^{fit}$ as a function of both $k_{ex}^{(1)}$ and $k_{ex}^{(2)}$ for $\theta^{(1)} = \theta^{(2)} = 70.5°$ (d) and $\theta^{(1)} = 20.0°$ and $\theta^{(2)} = 70.5°$ (e). The position of the one-dimensional slices shown in b and c are indicated by dashed lines.

the tilt angle between the inner $C_3$-axes and its own symmetry axis ($\theta^{(2)}$). As an example, the apparent recoupling behaviour for CP recoupling for different time scales of the inner and outer motion is shown in Fig. 10b-e. The fitted apparent $\delta_{IS}^{fit}$ for $\theta^{(1)} = \theta^{(2)} = 70.5°$ is shown in Figs. 10b and d, while Figs. 10c and e shows simulation results for an inner motion with a smaller amplitude ($\theta^{(1)} = 20.0°$ and $\theta^{(2)} = 70.5°$). When both motions are slow ($k_{ex} < 1 \cdot 10^2$ s$^{-1}$ for the $\delta_{IS}/(2\pi) = 5$ kHz considered here), the full interaction is observed. When both motions are fast on the other hand (ca. $k_{ex} > 1 \cdot 10^5$ s$^{-1}$), the scaled interaction is observed, where the total scaling factor corresponds to $P_2(\cos\theta^{(1)}) \cdot P_2(\cos\theta^{(2)})$. If the amplitude of the inner motion is small and the motion is sufficiently fast, the transition region for the outer motion is shifted (see Fig. 10e), since the inner motion already leads to a scaling of the dipolar coupling. The effects for motion on intermediate time scales depends on the amplitude of the two motions and the relative speed of the inner and outer motion and are difficult to predict in general.

## 4   Conclusions

We have investigated the averaging of anisotropic interactions in solid-state NMR under MAS using numerical simulations based on the stochastic Liouville equation. Simple jump models with a three-fold symmetry and equal populations were used to simplify the characterization of the partially averaged couplings using a single order parameter. In all cases, the time scale of the dynamics defines three distinct regions: slow motion where the full anisotropic interaction is retained, fast motion where a scaled anisotropic interaction is obtained and an intermediate region where a transition from the full to the scaled anisotropic interaction is observed. The time scales included in the three regions depend on the magnitude of the interaction and, to a much

lower extent, on the method used to measure the anisotropic quantity while the MAS frequency has a negligible influence.

Heteronuclear one-bond H-X dipolar couplings are the most often measured interactions for the characterization of the amplitude of motion (order parameter), and are often combined with relaxation studies. The position of the transition region depends on the magnitude of the dipolar coupling. For typical heteronuclear one-bond (e.g., NH, CH) dipolar couplings with an anisotropy of $\delta_{IS}/(2\pi)$ on the order of several 10 kHz, averaged interactions are observed for exchange-rate constants exceeding roughly $10^5$ s$^{-1}$ with minor differences between different recoupling methods. Smaller dipolar couplings shift the
transition region to slower time scales. A scaling of the effective dipolar couplings by pulsed recoupling methods only has a minor influence on the position of the transition but can influence the spectra obtained in the transition region strongly. The measurement of scaled CSA tensors behaves similarly to the dipolar couplings since it is based on the same principles. For a CSA tensor with an anisotropy of $\delta_{CSA}/(2\pi) = 5$ kHz, the scaled interaction is observed for $k_{ex} > 10^4$ s$^{-1}$ and is independent of the MAS frequency.


    The determination of dipolar couplings using off-magic angle spinning behaves differently from the other methods: the transition region starts at much slower rate constants (around 1 s$^{-1}$) and extends to roughly 1 s$^3$. However, for MAS frequencies up to 100 kHz the end of the transition region overlaps with the motional time scales for which efficient transverse relaxation is observed. Thus, the range of exchange-rate constants for which the anisotropy of the dipolar coupling is difficult to determine is extended to rate constants up to $10^7$ s$^{-1}$. For off-magic angle spinning, the transition region does not coincide with motional
time scales that cause rapid transverse relaxation. The extended dynamic time scales towards slower motions covered by off-magic angle spinning is mirrored by the large range of dynamics down to millisecond time scales obtained in residual dipolar couplings in partially aligned liquids (Blackledge, 2005). More detailed studies of the interplay between the scaling parameter and the strongly-broadened transition for off-magic angle spinning are are currently under way.


    Quadrupolar couplings (typically $^2$H or $^{14}$N) under MAS do not require active recoupling due to their typically larger magnitude. They can be measured directly using the side-band pattern of the first-order quadrupolar coupling. For exchange-rate constants larger than $10^7$ s$^{-1}$, scaled quadrupolar couplings are obtained while the full coupling is measured for exchange-rate constants smaller than $10^3$ s$^{-1}$. In the transition region ($k_{ex} \approx 10^3$ s$^{-1}$ to roughly $10^7$ s$^{-1}$) strong line broadening is observed that
obscures the side band pattern.

    Combining measurements of large anisotropic interactions (e.g., quadrupolar couplings) with measurements of intermediate (e.g., one-bond-heteronuclear dipolar couplings or CSA tensors) and small anisotropic interactions (e.g., off-magic angle spinning) might be a possibility to characterize the amplitude of motion in different time windows. However, care has to be taken
that all interactions probe the same set of motions. While such a combination of different experiments that are sensitive to anisotropic interactions could be a way to gain information on the time scales of motion, relaxation-based experiments appear

to be the better and more robust and reliable way of accessing time scales of dynamics.

*Code and data availability.* The simulation programs as well as all processing and plotting scripts are available at https://doi.org/10.3929/ethz-
b-000666765.

*Author contributions.* ME and PS designed the research, KA and LB carried out the simulations and data analysis. KA wrote the first draft of the manuscript with support from LB. All authors discussed the results and were involved in finalizing the manuscript.

*Competing interests.* At least one of the (co-)authors is a member of the editorial board of Magnetic Resonance. The authors have no other competing interests to declare.

*Acknowledgements.* We would like to thank Kay Saalwächter for pointing out important aspects of the intermediate regime during the open review process. L. M. B. is recipient of a DOC fellowship of the Austrian Academy of Sciences at the Institute of Science and Technology Austria (DOC-OEAW, PR10660EAW01). This research has been supported by the ETH Zürich and the Schweizerischer Nationalfonds zur Förderung der Wissenschaftlichen Forschung (grant nos. 200020_188988 and 200020_219375).

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
