# Peer review of "Evaluating the motional time scales contributing to averaged anisotropic interactions in MAS solid-state NMR"

_Magnetic Resonance, 2024_

## Author Response (AR1)

Ms. No.: MR-2024-4
Title: Dynamic averaging of anisotropic interactions and its dependence on motional time scales in MAS solid-state NMR
Corresponding Author: Dr. Matthias Ernst
Authors: Kathrin Aebischer

**Response to reviewers**

Firstly, we would like to thank all reviewers for their valuable feedback and suggestions to improve the presentation of our work. While most points have been addressed in the direct responses to the comments made by the three reviewers, we would like to provide a summary of the response and the modifications implemented in the manuscript in the following. A marked-up manuscript showing all the changes can be found at the end of this letter.

**Reviewer 1:**

The main concern of Reviewer 1 was the missing references and discussion of approaches to determine motions on the intermediate time scale which we have discussed extensively online. Through this criticism, we realized that we were not entirely successful in conveying the main focus of our work. The aim of this paper was to elucidate which dynamic time scales contribute to partially averaged interactions that are often measured in combination with relaxation data to improve the characterization of the amplitude and time scale of motion. Moreover, we wanted to study if and how this depends on the method used and the experimental parameters such as the MAS frequency. Furthermore, we agree that a discussion of methods to characterize motion in the intermediate time scale would significantly improve the paper. To address these issues, we have made several changes to the manuscript:

- We changed the title to *"Evaluating the motional time scales contributing to averaged anisotropic interactions in MAS solid-state NMR"* to emphasize the main focus of the paper. We believe that the new title, together with the abstract conveys the topic of the paper well.
- Throughout the text, we have tried to put more emphasis on the fast dynamics time scale and less on the intermediate time scale.
- A new paragraph (on page 3/4) has been included in the introduction where we discuss the challenges associated with characterizing motion on intermediate time scales. In particular, we discuss how relaxation affects the different recoupling experiments as well as other steps in the experiment (e.g. detection period, polarization transfer steps). We reference relevant examples from the literature where intermediate time scale motions have been characterized.
- In the Methods section (page 6), we reference the more advanced data analysis techniques reported in the literature for characterizing motion on intermediate time scales that were suggested by the reviewer. We discuss in more detail the limitations of our simple data fitting approach. Nevertheless, we still believe that the simple fitting procedure we have adopted in the manuscript is sufficient for the characterization of the fast motion regime for our noise-free simulation data.

- In the description of the wPARS experiment (page 8), we mention the possibility to compensate also such recoupling experiments with a reference experiment and give a literature reference to such an implementation.
- Responding to Reviewer 1's suggestions, we incorporated a discussion on T1rho/T2 relaxation in the results and discussion section (pages 12-14) of the dipolar recoupling simulations. We explain the differences in the relaxation (and its compensation) for the recoupling sequences treated in this work. Moreover, we added a figure (see new Fig. 5 in the main text) characterizing the signal decay during the REDOR reference experiment (S0 curve) to show how the observed signal decay depends on the coupling strength and MAS frequency.
- In the discussion of the quadrupolar side-band spectra (page 20), we mention again the fact that the line broadening is side-band dependent and give a reference to the relevant papers.
- We have reworked and shortened the conclusion to only include statements concerning the main focus of the paper.

A second request was a more detailed discussion of the off-angle experiment and its sensitivity to motions on different time scales. We agree that this is an interesting topic, but a more detailed investigation is beyond the scope of this paper and would require an experimental implementation of such off-angle experiments that can be used with fast MAS. We are currently working on this problem and hope to be able to address these questions in more detail in the future.

**Reviewer 2:**

Besides the above discussed changes in response to reviewer 1, the main concern of Reviewer 2 was that the code to generate spectra and the data evaluation should be made available. All of these data will be deposited in the ETH Library Data Repository where they are available with a DOI and the reference will be inserted in the final version of the paper.

**Reviewer 3:**

The main question was: "I wonder if the minimal effect on the wPARS sequence upon changing the window duration is because the addition of a window does not really change the effective Hamiltonian description (except for the scaling factor), while this is not true for the epsilon-REDOR and the tilted-angle CP experiment. If this is true, can one then expect that as one spins faster, the deviations that one sees in CP and REDOR will progressively get smaller?"
We believe that the minimal effects in the wPARS sequence come from the relatively small changes in the magnitude of the effective Hamiltonian compared to the other sequences.

We hope that these changes address the concerns raised by the reviewers and believe that the discussion of the relaxation effects for intermediate time scale motion improves the manuscript.

---

## Author Response (AR2)

Ms. No.: MR-2024-4
Title: Dynamic averaging of anisotropic interactions and its dependence on motional time scales in MAS solid-state NMR
Corresponding Author: Dr. Matthias Ernst
Authors: Kathrin Aebischer

**Response to reviewers**

**Reviewer 1:**

The authors have provided a revised version that finds my appreciation in all but one aspect. The reviewer's own previous work, arguably constituting the state of the art for several aspects of the current work, has not been referred to: Saalwächter and Fischbach, J. Magn. Reson.157 (2002) 17. Its results should be discussed in the appropriate places. Our work was the first to provide numerical simulation results of the effect of the motional regime transition from slow to fast for the results of MAS recoupling experiments, using a simulation strategy that resembles the authors' code in many regards (we also ignored off-diagonal terms in the Liouvillian, and further reduced storage space and calculation time by using a separate linear transformation for the exchange dynamics). The experiment of interest was REDOR (same as in the manuscript), but applied to CSA-recoupling (along the lines of the CODEX experiment). We did explore the measurement of the important T2 effect upon recoupling, and our results for the apparent fitted interaction strength closely match those of the authors (compare our Fig.6 with Fig.3 of the manuscript). The main difference is that we did not explore the effect of variable MAS frequency – on the other hand, we confirmed our results also by experiments.

**Response:**
We apologize for the oversight of this paper which got lost in the considerable rewriting of the intermediate exchange regime. We have added a reference to this paper which indeed implements very similar numerical simulations. We added a sentence to the introduction where we mention the previous implementation. We also added a sentence to the methods section where we again say that similar but simplified methods were implemented before. We hope that these changes make the paper now acceptable for publication.